# Spatial Differences of Nutrient Adequacy in Coastal Areas of China

**DOI:** 10.3390/nu14224763

**Published:** 2022-11-11

**Authors:** Wei Yin, Huijuan Yu, Yecheng Wang, Rongshan Qiu, Limin Han

**Affiliations:** 1Management College, Ocean University of China, Qingdao 266100, China; 2Institute of Marine Development, Ocean University of China, Qingdao 266100, China

**Keywords:** nutrient adequacy, nutrition security, spatial differences, coastal areas, production and consumption, China

## Abstract

Ensuring nutrient adequacy for all is a common goal of the international community, but spatial difference is one of the barriers to its development. Exploring nutrient adequacy in coastal areas of China can help regions where food production systems and economic development systems are under mutual stress to reduce nutritional disparities and improve nutrition levels. This paper used the transformation food-to-nutrient model to calculate nutrient production and nutrient consumption in 11 coastal provinces of China and analyzed their spatial patterns, after which spatial differences in nutrient adequacy (including energy, protein and fat) were analyzed. The results showed that nutrient production and nutrient consumption in coastal areas of China showed significant spatial differences, in which nutrient production was mainly concentrated in land food, and the three provinces of Shandong, Jiangsu and Hebei contributed more. Guangdong had the highest nutrient consumption; in contrast, Shanghai, Tianjin, and Hainan had the lowest consumption. Nutrient adequacy was not optimistic, with fat being particularly significant, and nutrient surplus quantity was mainly concentrated in Shandong and Jiangsu and nutrient deficiency quantity was mainly concentrated in Guangdong. Overall, the study area had adequate levels of protein and was deficient in energy and fat levels, with surplus or shortage of 2.41 million tonnes, 2620 billion kcal and 9.97 million tonnes, respectively.

## 1. Introduction

Ensuring healthy diets and nutrient adequacy for everyone is a topic of global concern [1,2]. The international community is highly concerned about food and nutrition security, has gathered a great deal of consensus [3], and has set specific development goals, such as the Sustainable Development Goal 2 (SDG-2) to “end hunger, achieve food security and improved nutrition and promote sustainable agriculture” [4], and the UN Decade of Action on Nutrition. However, between 720 million and 811 million people worldwide still face hunger, 2.37 billion people do not have access to adequate food, and nearly 3.1 billion people cannot afford a healthy diet in 2020 [5,6]. Nowadays, it has become more difficult to achieve the above goals and programs, due to the impact of multiple crises such as climate change, regional conflicts, and the COVID-19 epidemic [5].

As the most populous developing country in the world, it is undeniable that China faces numerous and complex challenges in feeding and nourishing 18 percent of the world’s population with only 9 percent of the world’s arable land and 6 percent of the world’s fresh water [7,8]. Despite this, it has achieved unprecedented and miraculous achievements in reducing domestic poverty and hunger in the past decades. China’s coastal areas, like the global coastal areas generally, are highly populated and economically developed [9], with the 11 coastal provinces accounting for 13.60% of the country’s land area, containing 45.05% of its population and generating 53.15% of its GDP (2020 data). These areas are the frontier areas of China’s economic development and foreign trade, playing a key role in radiating and driving inland development [10].

Unlike the economic status of China’s coastal areas, the food nutrition supply status is not encouraging [11], as the food supply faces the twin pressures of low potential to increase production and still-growing food consumption. In terms of food production, due to the rapid urbanization process in coastal areas, a large number of agricultural production factors, such as arable land, water, and labor force, have been transferred to industry and service, and the space left for food production has been gradually compressed [12,13]. With the gradual increase in pressure on ecological and environmental protection, the model of maintaining high output with high input factors (chemical fertilizers, pesticides, water, land, etc.) is being gradually eliminated, and the way of increasing food production is gradually shifted to a large-scale, intensive and sustainable production model [14,15], but the potential of production increasing cannot be determined and remains to be further observed. In terms of food consumption, Chinese residents are in a period of food consumption transformation and upgrading, and this phenomenon is more obvious in coastal areas and is mainly reflected in the diversification of diet structure and the significant increase in per capita consumption of meat and dairy products [16,17]. This transformation and upgrading of the food consumption structure means that more first-trophic-level foods such as grains and legumes need to be converted to second-trophic-level foods such as meat, which leads to a further increase in the total food nutrient requirements [18,19].

Free trade and the frequent flow of food (possibly domestically and internationally) could allow areas of high economic consumption power to solve the problem of local food production being unable to meet the consumption through purchasing supplies, ensuring the minimum bottom line of nutrient adequacy still has strategic value for the region to deal with emergencies and ensure societal stability [20]. At the same time, it important to also conduct a self-review of regional nutrient production and consumption.

Food security has always been a key concern for the Chinese government and academia [21,22]. Additionally, their focus has long been on food production capacity enhancement [23,24], production efficiency enhancement [25,26] and analysis of the factors influencing food production [15,27,28]. However, nutrition issues have been less integrated into the framework of agricultural policy analysis and agricultural development, resulting in a lack of clarity about the nutritional supply status of Chinese people and a disconnect between agricultural production and nutritional intake [29,30,31]. Nutrient adequacy, which is defined as whether nutrient production can meet nutrient consumption by people, can be used as an indicator to measure the nutritional status of a country or region [32,33]. In China, it is an innovative work to evaluate food and nutrition security using nutrient adequacy (nutrition-oriented indicator) instead of per capita food ownership (yield-oriented indicator). At the same time, scholars have mostly conducted nutritional status assessments at the national level [4,33], but these macroscopic studies cannot fully explain the changing characteristics and spatial differences in nutrient production and consumption in coastal areas of China. The existing studies also have some irrationality in data selection. For example, seafood plays a very important role in dietary nutrition for coastal residents [34], but is often excluded from the food nutrition system in nutrient production calculation practices [32]. When accounting for nutrient consumption, CHNS data (only covering less than half of China’s provinces) or 24 h (or 48 h) dietary recall questionnaires in small areas are often used [35,36], but none of these data can be applied effectively to China’s 11 coastal provinces.

This study helps to provide advice to the Chinese government on how to increase nutritional self-sufficiency in coastal areas and reduce regional nutrient adequacy gaps. This study aimed to (1) calculate the nutrient production in coastal areas of China using food yield data and China food composition tables, and analyze the spatial differences it presents; (2) calculate the nutrient consumption based on food consumption data and China food composition tables, and analyze the spatial differences it presents; (3) evaluate the nutrient adequacy and spatial differences using the results of nutrient production and nutrient consumption.

## 2. Data and Methods

### 2.1. Research Areas

This paper selected 11 coastal provinces of Liaoning, Hebei, Tianjin, Shandong, Jiangsu, Shanghai, Zhejiang, Fujian, Guangdong, Guangxi and Hainan in China (Figure 1). The above-mentioned areas have a land area of more than 1.3 million km^2^, accounting for about 13.60% of the total land area of the country, and more than 3 million km^2^ of sea area for use; a population of 635.2 million, accounting for 45.05% of the national population; and a GDP of 53.67 trillion RMB, accounting for 53.15% of the total national GDP.

### 2.2. Transformation Model of Nutrient Production

Agriculture produces many types of food, but the nutrition composition of foods is different. If we use the cumulative weight of output to measure food production capacity, the nutrition differences between foods will be ignored, leading to inaccurate evaluation results. Therefore, the nutrition transformation approach is used to convert multiple foods into nutrients of uniform unit [11], which shows the advantages of unit consistency and comparability of attributes. The paper uses the two basic data of a food yield and food composition table to calculate the nutrient production, which is completed by multiplying food yield and nutrients per unit of food. Considering the availability and completeness of data, we choose the three most important nutrient indicators, i.e., energy, protein (protein being derived from various types of food, and therefore, including not only plant-based proteins, but also animal-based proteins), and fat. The formula is as follows:(1)$$NP_{i,j}=\sum_{k=1}^{13}FY_{i,k}\times EP_{k}\times NPUF_{j,k}$$
where *i* represents 11 provinces in China’s coastal areas; *j* represents nutrients, including energy, protein and fat; *k* represents the main types of food consumed by Chinese residents on a daily basis, i.e., grain, edible oil, vegetables, pork, beef, mutton, other meat, poultry, aquatic products, eggs, milk, fruits, and sugar; *NP* represents the nutrient production; *FY* represents the food yield; *EP* represents the edible proportion; *NPUF* represents the nutrient per unit of food.

The following two aspects were fully considered in the calculation of food nutrient production to ensure that the results are scientific and accurate.

On the one hand, reserving seeds for food is a common phenomenon in agricultural cultivation. For this study, therefore, the rate of reserving seeds for certain foods was set at 2% [37]. Food loss also refers to the decline in food quality and quantity in the process of storing, transporting, and processing due to objective technical limitations and the considerations of economic costs [38]. Therefore, this part of food loss should not be included in the total food nutrient production. Yin et al. [39] clearly discusses the rate of food loss in China. Their results are representative, so we drew on them to determine the loss rate of various types of food in China. Due to the lack of the rate of food loss at the provincial level, the paper adjusted this parameter with the help of the agricultural input funds per unit of grain output (a significant negative correlation with food loss rate [40]), using the fixed scale scaling method. The formula is as follows:(2)$$FY{\left(1\right)}_{i,k}=IFY_{i,k}\times \left(1-PRS_{k}\right)$$
(3)$$FY{\left(2\right)}_{i,k}=FY{\left(1\right)}_{i,k}\times \left(1-FLR_{i,k}\right)$$
where *IFY* represents initial food yield, i.e., official statistics; *PRS* represents proportion reserved for seeds; *FY*(1) is food yield after subtracting seeds; *FLR* represents food loss rate; *FY*(2) is food yield after subtracting seeds and losses.

On the other hand, production of animal food needs to consume a large amount of plant-based feed grains in the feeding process, so the nutrition provided by this portion of feed grains should not be counted in the total nutrient production. There are two core parameters in the evaluation of feed grains, namely feed conversion ratio and feed formulation. Due to the differences in production level, technology and mode, the two core parameters above also show regional differences. For this reason, we cited research results in specialized fields to determine the feed conversion ratio parameter data [41], as well as the proportion of animal food feed formula [42] of 11 coastal provinces, so as to calculate the amount of feed grains consumed. The formula is as follows:(4)$$QFD_{i,k\left(1\right)}=\sum_{n=1}^{8}MY_{i,n}\times DPA_{n}\times FCR_{i,n}\times FF_{n,k\left(1\right)}$$
(5)$$FY_{i,k}=FY{\left(2\right)}_{i,k}-QFD_{i,k\left(1\right)}$$
where *n* represents the types of animal food, including pork, beef, mutton, other meat, poultry, aquatic products, eggs, and milk; *k*(1) represents the types of feed grains, *k*(1) ∈ *k*; *QFD* represents quantity of feed grains; *MY* represents meat yield (live); *DPA* represents dressing percentage of animals; *FCR* represents feed conversion ratio; *FF* represents feed formula; *FY* is the final data of food yield used in this study.

### 2.3. Transformation Model of Nutrient Consumption

There are many different types of edible foods for humans, and different foods have different nutrient contents. Therefore, it is necessary to convert different foods into nutrients of uniform unit to facilitate comparison [43]. Nutrient consumption is calculated by multiplying per capita food consumption, nutrient per unit of food and population. Eating out is not included in the official statistics on food consumption per capita in China. The frequency of eating out, however, has increased with the rapid increase in Chinese incomes over the past several years [44]. The omission of these data will lead to a large deviation in the assessment results of food nutrient consumption. This article refers to the rate of food away from home in targeted research results [45] and uses the per capita consumption expenditure data (a significant positive correlation with the rate of food away from home [46]) to adjust the rate of food away from home in coastal provinces. In early 2020, due to the COVID-19 outbreak, China introduced strict long-term home quarantine measures. Thus, this paper multiplied the rate of food away from home by half in each province in 2020. The formula is as follows:(6)$$NC_{i,j}=\sum_{k=1}^{13}PCFC_{i,k}\times \left(1+FAFH\right)\times NPUF_{j,k}\times Pop_{i}$$
where *NC* represents nutrient consumption; *PCFC* represents per capita food consumption; *PAFH* represents the rate of food away from home; *NPUF* represents the nutrient per unit of food; *Pop* represents population.

### 2.4. Nutrient Adequacy

Nutrient adequacy is defined as an indicator to measure whether nutrient production can effectively meet nutrient consumption in this study, and is the ratio of nutrient production to nutrient consumption [32,33]. The theoretical value range of the ratio is (0, +∞). When the ratio is between 0 and 1, it means that nutrient production cannot meet consumption, and the closer the value is to 0, the worse this deficiency is; when the ratio is 1, it means that nutrient production only just meets consumption; when the ratio is greater than 1, it means nutrient production is sufficient to meet consumption; the further the value away from 1, the more significant this adequacy is. The formula is as follows:(7)$$NA_{ij}=\frac{NP_{ij}}{NC_{ij}}$$
where *NA* represents nutrient adequacy; *NP* represents nutrient production; *NC* represents nutrient consumption.

Moreover, in order to further investigate the adequacy of nutrients in coastal areas of China, we introduced the indicator of nutrient surplus or deficiency quantity to indicate the size of regional nutrient surplus or deficiency in order to help us better understand the regional food nutrient production and consumption. The formula is as follows:(8)$$NSDQ_{j}=NP_{j}-NC_{j}$$
where *NSDQ* represents nutrient surplus or deficiency quantity. When *NA* is greater than 1, *NSDQ* is greater than 0, and it represents the status of nutrient surplus quantity (*NSQ*); when *NA* is less than 1, *NSDQ* is less than 0, and it represents the status of nutrient deficiency quantity (*NDQ*).

### 2.5. Data

The data of food yield, per capita food consumption and population used for our study come from the China Rural Statistical Yearbook, China Fishery Statistical Yearbook and the website of National Bureau of Statistics of China (http://data.stats.gov.cn/, accessed on 17 June 2022). Data on food nutrition composition comes from China Food Composition Tables compiled by the Chinese Nutrition Society. Spatial differences in nutrient production, nutrient consumption, and nutrient adequacy are the focus of this paper, which also considers the availability, comparability and completeness of the data, and so we chose the sample data from 2015 to 2020. Furthermore, the average values from 2015 to 2020 were used to complete the quantification of specific indicators for China’s coastal areas.

## 3. Results

### 3.1. Nutrient Production in Coastal Areas of China

In terms of the nutrient production of seafood (Table 1), China’s 11 coastal provinces can be divided into two categories. The first category is the provinces with larger production, i.e., Fujian, Shandong, Zhejiang, and Guangdong, whose energy production ranges from 2410 billion to 3560 billion kcal, protein production ranges from 336,700 to 437,900 tonnes, and fat production ranges from 59,900 to 83,200 tonnes. The second category is the provinces with smaller production, namely Liaoning, Guangxi, Hainan, Jiangsu, Hebei, Tianjin, and Shanghai. These provinces produce less than 1430 billion kcal of energy, less than 204,700 tonnes of protein, and less than 26,400 tonnes of fat.

In terms of nutrient production of land food (Table 1), China’s 11 coastal provinces can be divided into three categories. The first category consists of Shandong, Jiangsu and Hebei, where the energy production exceeds 75,300 billion kcal, the protein production exceeds 2.637 million tonnes, and the fat production exceeds 999,100 tonnes. The second category consists of the provinces with lower nutrient production, namely Guangxi, Liaoning and Guangdong, whose energy production ranges from 29,830 billion to 55,850 billion kcal, protein production ranges from 746,800 to 1,164,000 tonnes, and fat production ranges from 550,500 to 807,700 tonnes. The third category consists of provinces with the lowest production, namely Zhejiang, Fujian, Hainan, Tianjin, and Shanghai, whose energy production is less than 21,010 billion kcal, protein production is less than 654,900 tonnes, and fat production is under 296,000 tonnes.

In terms of quantitative rank and spatial pattern, the total nutrient production in coastal areas of China is basically similar to that of land areas. Shandong and Jiangsu have the highest energy and protein production; Shandong and Hebei have the highest fat production. By thoroughly analyzing the above results, we innovatively identify two unique regional characteristics of nutrient production in coastal areas of China. One is that the contribution of seafood to the composition of nutrient production in coastal areas is extremely low (3.22~13.34%), while the contribution of land food is high (86.67~96.78%), and the contribution of nutrient production is mainly concentrated in three provinces, i.e., Shandong, Jiangsu, and Hebei. This indicates that the nutrient production in China’s coastal areas is strongly dependent not only on land food, but also on major grain-producing areas (Shandong, Jiangsu and Hebei are the major grain producing areas in China).

The other characteristic is inconsistency in the ranking of production between land and sea, and inconsistency in the ranking of production among nutrient types in China’s coastal areas. For example, Fujian ranks high (1st) among the 11 coastal provinces in terms of energy, protein, and fat production of seafood, but has a lower ranking in terms of nutrient production of land food (8th in energy, 11th in protein, 8th in fat). The inconsistency in the nutrient production ranking of land food and seafood is mainly caused by the difference in natural resource endowment of land and sea. That is to say, Fujian ranks 2nd among coastal provinces in terms of coastline length, but 8th in terms of sown area of farm crops. Another example is that Liaoning ranks 6th and 7th among the 11 coastal provinces in total energy and protein production, but ranks 4th in fat production. The inconsistency in the nutrient production ranking is mainly caused by the difference in the structure of food production. That is, Liaoning has higher relative production levels of fat-rich beans, oil crops and meat.

### 3.2. Nutrient Consumption in Coastal Areas of China

Figure 2 shows spatial differences in food nutrient consumption in coastal areas of China. In terms of energy consumption, Guangdong has the highest level of consumption, with 100,880 billion kcal consumption; Shandong, Jiangsu, Hebei, and Zhejiang consume more than 50,000 billion kcal; these are followed by Guangxi, Liaoning, and Fujian, whose consumption ranges from 32,320 to 37,860 billion kcal; while Shanghai, Tianjin, and Hainan, less than 18,950 billion kcal are consumed. In terms of protein consumption, Guangdong still has the highest consumption at 3,590,600 tonnes, followed by Shandong (2,493,800 tonnes). Consumption in the six provinces of Jiangsu, Hebei, Zhejiang, Guangxi, Liaoning, and Fujian ranges from 1,074,400 to 1,848,500 tonnes, while Shanghai, Tianjin and Hainan range for less than 663,100 tonnes. In terms of fat consumption, Guangdong is much higher than the second with 4,067,700 tonnes, followed by Shandong, Jiangsu, Zhejiang, Hebei, Guangxi, Liaoning, and Fujian, with consumption range from 1,196,100 to 2,336,900 tonnes, while Shanghai, Tianjin, and Hainan consume less than 716,300 tonnes.

There is a certain regional gap in the daily food nutrient consumption per person in the coastal areas of China, which reflects the difference in food consumption structure among regions. Using the data on total food nutrient consumption in coastal areas, we made further calculations to obtain the daily food nutrient consumption per person. The results in Table 2 show that Zhejiang has the highest daily food energy consumption per person with 2331 kcal and Shandong has the lowest with 1929 kcal. In Guangdong, daily food protein and fat consumption per person is the highest with 81 g and 91 g, respectively, while the lowest consumption of protein is in Jiangsu with 60 g and the lowest consumption of fat is in Hebei with 63 g. By restoring and comparing the per capita food consumption data for the two more typical provinces of Hebei and Guangdong, it can be seen that the annual per capita consumption of pork, poultry and aquatic products in Hebei is 13.3 kg, 5.5 kg and 6.7 kg, respectively, while the consumption of these three food items in Guangdong is 29.8 kg, 22.9 kg and 24.6 kg, respectively, which is significantly larger than the level in Hebei. This difference in food consumption structure results in a gap in the level of nutrient consumption. Despite the above data not being able to effectively assess the proportion of over- and under- nutrient intake of the population in each region, especially the nutritional status of low-income groups, the regional imbalance of nutrient consumption still raises concerns.

### 3.3. Spatial Differences of Nutrient Adequacy in Coastal Areas of China

Figure 3 shows spatial differences in food nutritional adequacy in the coastal areas of China. From the results of nutrient adequacy, there are four provinces with energy adequacy values greater than 1, namely Shandong, Jiangsu, Guangxi and Hebei; there are five provinces with protein adequacy values greater than 1, namely Jiangsu, Shandong, Hebei, Hainan and Guangxi; and there is no province with fat adequacy values greater than 1. This indicates that the self-sufficiency of food nutrition in China’s coastal provinces is very unpromising and that there is a problem of unbalanced nutrition supply, that more than half of the provinces do not produce enough nutrient for local consumption, and among them, the capacity of fat supply is the worst. Regarding specific values, the situation of energy supply in Shandong, Jiangsu, and Guangxi is relatively optimistic, with the values of energy adequacy exceeding 1.5. On the contrary, the energy supply in Shanghai, Fujian, Guangdong, Tianjin, and Zhejiang is not optimistic, with values of energy adequacy less than 0.5. The protein supply in Jiangsu and Shandong is relatively optimistic, with the values of protein adequacy exceeding 1.8. However, the protein supply in Shanghai, Guangdong, Tianjin, and Fujian is highly unpromising, with values of protein adequacy less than 0.5. None of the provinces has a fat adequacy value above 1, even Shandong, and the fat supply in Shanghai, Guangdong, Zhejiang, Tianjin, Fujian and Jiangsu is extremely poor, with fat adequacy values below 0.5 and even below 0.3 in many provinces.

The nutrient surplus in China’s coastal areas is mainly concentrated in Shandong and Jiangsu, while the nutrient deficiency is mainly focused in Guangdong. Shandong and Jiangsu provide 54,740 billion kcal and 45,810 billion kcal of energy surplus, respectively, accounting for 38.80% and 32.47% of the total energy surplus in the coastal areas. Two provinces also provide 2.21 million tonnes and 1.75 million tonnes of protein surplus, which accounts for 44.99% and 35.70% of the total protein surplus in the coastal areas. Neither of these two provinces has a fat surplus. In Guangdong, the energy deficiency quantity, protein deficiency quantity and fat deficiency quantity reach 64,770 billion kcal, 2.51 million tonnes and 3.46 million tonnes, respectively, accounting for 45.07%, 53.69% and 34.67% of the total deficiency quantity in coastal areas. Geographic agglomeration characteristics are reflected in nutrient surpluses and nutrient deficits in coastal areas of China. On the one hand, this spatial agglomeration reflects the extremes of regional gap in nutrient adequacy. On the other hand, regional concentration reduces complexity and improves the targeting of regional food nutrition supply capacity, which is more conducive to the formulation and implementation of policies.

Overall, the total amount of food protein in China’s coastal areas is sufficient, with a surplus of 2.41 million tonnes, while the total amount of energy and fat is insufficient, with a deficit of 2620 billion kcal and 9.97 million tonnes, respectively. This shows that China’s coastal areas produce almost enough protein to meet consumption, but face a shortage of energy and fat, especially fat. China is the most populous developing country in the world, and the Chinese government attaches great importance to the self-sufficiency rate of domestic cereals, believing that complete self-sufficiency of cereals is the basic way to eliminate hunger and to meet the energy needs of every person. However, unfortunately, as the results of this paper show, the amount of energy produced in the coastal areas of China still cannot meet the consumption of the local people and needs to be supplemented by food imported from abroad.

## 4. Discussion

Economic development and food production are both important. Most of China’s coastal areas are developed regions, where agricultural production resources are heavily encroached upon by the second and tertiary industries [13,47,48]. In order to promote economic development, local governments have taken up large amounts of natural resources to develop industries with high economic value, and either abandoned food production or produced small amounts of food with lower economic value. As a result, local residents rely heavily on the inflow of food from abroad, thus losing their autonomy in terms of food and nutritional security, which is not conducive to maintaining the stability of regional food markets. The Chinese government is also aware of this problem and has begun to gradually put pressure on local governments in economically developed regions to increase food self-sufficiency in these regions. Obviously, regions with high levels of economic development can improve food yields and production efficiency by investing more in innovation in agricultural technology and management models and adopting efficient and intelligent production methods to drive up food and nutrition production [49].

Sustainability of agri-food systems is a consideration in ensuring food and nutrition safety. The agri-food system in coastal China is in a vulnerable position (compromised by economic development) and faces serious negative challenges to its sustainable development, such as land, water and ecosystems degradation; climate change; biodiversity loss and the COVID-19 epidemic and other impacts on agricultural production [50,51], which means that the sustainability transition of agri-food systems should be an urgent issue to be addressed nowadays. To ensure food and nutrition security for the present generation without threatening the productive capacity of future agri-food systems, we can work together to improve food and nutrition security and agri-food system sustainability through different strategies, such as efficiency increase (e.g., sustainable intensification), demand restraint (e.g., sustainable diets), and food systems transformation (e.g., alternative food systems) [52]. In recent years, the Chinese government has introduced a large number of policies to improve the agro-ecological environment and enhance the sustainability of the agri-food system, which have achieved remarkable results. These initiatives include the action of reducing chemical fertilizer and pesticide use [53], the grain for green project [54], the black soil conservation project, and the action to reduce food loss and waste [55].

Population is a key factor in determining nutrient consumption in China’s coastal areas. Guangdong > Shandong > Jiangsu > Hebei > Zhejiang > Guangxi > Liaoning > Fujian > Shanghai > Tianjin > Hainan is the order of food nutrient consumption, and so is the population order in China’s coastal areas. In other words, with the continuous population growth of China (in fact, the population growth rate of China has decreased) and the continuous upgrading of food consumption structure (advanced nutrition consumption), it can be expected that China’s food nutrient consumption will continue to increase for some time [24]. It is a paradox that fat is serious shortage in coastal China, while obesity rates are widespread and rising [56]. The reasons for this contradiction may include, but are not limited to, regional inequalities in fat consumption due to differences in economic development, the supplementation of large amounts of non-local foods (including foreign food), the irrational dietary structure in households, and irrational intake of carbohydrates [39,57].

Reducing regional gaps in nutrient adequacy is a critical component to achieving the goal of healthy diets for everyone. Global food production is sufficient to feed the world population [58], but due to the long-term food production and consumption of different countries varying widely, food ownership and nutrient intake also vary widely. Regional differences exist objectively. The most fundamental reason for the differences is the different natural resource endowments of regional food production, followed by the different intensity of regional inputs to food production elements under different levels of economic development. Even though food trade has compensated for food shortages and nutritional imbalances in some regions [59], huge gaps in nutrient intake still exist between regions. Undeniably, reducing regional gaps in nutrient adequacy is a challenging task, because it is related to a wide range of regions, a large population and many stakeholders, and it is also profoundly influenced by so many factors such as trade security, regional economic strength, population consumption levels, food welfare and national development policies. Besides fully exploiting regional production capacity, the key to regional nutrient adequacy is to actively engage in food cooperation with international organizations and other regions. China’s approach has been to provide substantial support to poor areas in terms of policy, funding and human capital, and to focus on the livelihoods of people living in poor areas [60,61].

High-quality protein can be obtained from seafood, and seafood resources can be further explored. Although protein is abundant in coastal areas of China, according to Yin’s research results [39], the protein composition of Chinese food output is still dominated by a large proportion of plant-based proteins, while the proportion of animal proteins with high quality is not high, and there is an unreasonable protein production structure problem. Although there is a large gap between seafood and land food in terms of total nutrient production, seafood can produce animal protein with a lower environmental burden and has a promising future [34,62,63], and therefore contributes not only to improved nutrition and sustainable consumption, but also to reduced production and ecological stress on land food. China’s coastal areas can try to utilize offshore water resources to develop large-scale offshore green aquaculture to improve the efficiency and technology of farming and meet the consumer demand for high-quality protein. The Chinese government has been supporting the enhancement of seafood production capacity and is currently promoting the construction of ocean ranching and offshore farming with preferential policies in terms of approval authority and financial subsidies [64,65].

Our research has several limitations. Due to data availability, only three nutrients were selected for this study: energy, protein, and fat. Many other nutrients are still very important, such as carbohydrates, vitamins, and minerals, but were not included in the study. We encourage more scholars to participate in further research on the adequacy, balance and safety of more nutrients. This study focuses on spatial differences in nutrient adequacy, but neglects the nutritional status of different population groups, which is necessary and meaningful, especially neglecting to focus on low-income populations, populations in war-torn areas, women and children, etc. We strongly appeal to local governments, international organizations, and social groups to make efforts to help more people escape from nutrition insecurity.

## 5. Conclusions

China’s coastal areas have high population density and economic density, which is a typical area of mutual stress between food production system and economic development system. In this paper, we measured the production and consumption of nutrients (energy, protein and fat) in 11 coastal provinces of China using data from 2015 to 2020, and completed a mapping (self-review) of the production and consumption of nutrients in coastal areas. Based on this, nutrient adequacy was used to explore the nutritional status and spatial differences, providing policymakers with reference information and implications to reduce regional nutritional disparities and improve nutritional levels. Important conclusions from our results include the following:

First, in coastal areas of China, nutrient production is greater in the land areas of Fujian, Shandong, Zhejiang and Guangdong, and in the sea areas of Shandong, Jiangsu and Hebei. In the total nutrient production, the contribution of seafood is very small (3.22~13.34%) and the contribution of land food is large (86.67~96.78%), moreover, provinces with high contribution to nutrient production are mainly concentrated in three provinces, Shandong, Jiangsu and Hebei.

Second, Guangdong has the highest nutrient consumption, its energy, protein and fat consumption are 100,880 billion kcal, 3,590,600 tonnes, 4,067,800 tonnes, respectively. In contrast, Shanghai, Tianjin, and Hainan have the lowest consumption. At the same time, there is a certain degree of regional disparity in daily food nutrient consumption per person in coastal areas, which reflects the spatial differences in food consumption structure.

Finally, nutrient adequacy in China’s coastal provinces is not optimistic and superimposed on the problem of nutritional imbalance, with more than half of the provinces not producing enough nutrient for local consumption, and the capacity for fat supply being very poor. Shandong and Jiangsu are the main sources of nutrient surplus, and Guangdong is the main source of nutrient deficiency. The total amount of protein is sufficient, and the total amount of energy and fat is insufficient, and the fat deficiency is especially most obvious.

## Figures and Tables

**Figure 1 nutrients-14-04763-f001:**
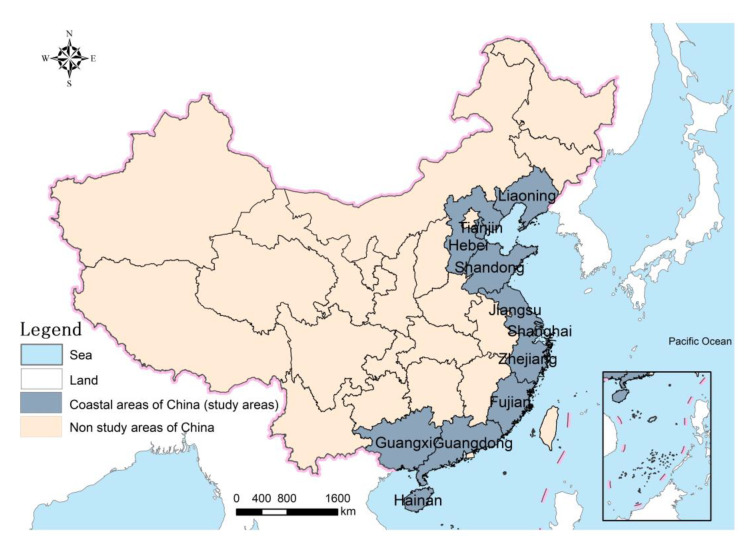
Location map of China’s coastal areas.

**Figure 2 nutrients-14-04763-f002:**
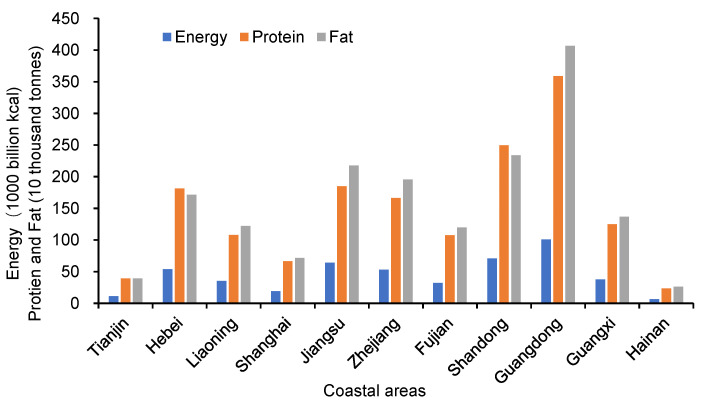
Nutrient consumption in coastal areas of China.

**Figure 3 nutrients-14-04763-f003:**
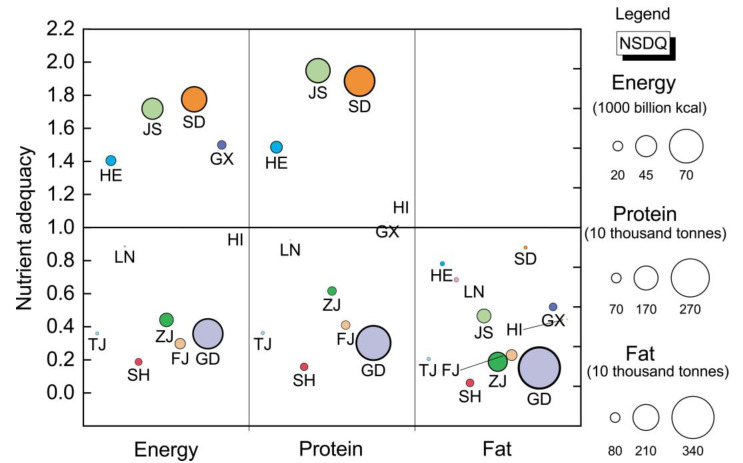
Spatial differences of nutrient adequacy in coastal areas of China.

**Table 1 nutrients-14-04763-t001:** Land and seafood nutrient production in coastal areas of China.

Area		Energy (1000 Billion Kcal)			Protein (10 Thousand Tonnes)			Fat (10 Thousand Tonnes)		
		Sea Food	Land Food	Total	Sea Food	Land Food	Total	Sea Food	Land Food	Total
Tianjin	TJ	0.03	3.91	3.94	0.43	13.69	14.12	0.08	8.01	8.09
Hebei	HE	0.23	75.30	75.54	4.04	263.70	267.73	0.46	133.47	133.93
Liaoning	LN	1.42	29.83	31.25	20.46	79.61	100.08	2.63	80.69	83.32
Shanghai	SH	0.01	3.55	3.56	0.12	10.33	10.45	0.02	4.36	4.38
Jiangsu	JS	0.65	109.19	109.84	9.77	350.37	360.14	1.51	99.91	101.43
Zhejiang	ZJ	2.41	21.00	23.41	36.92	65.48	102.40	7.10	29.59	36.69
Fujian	FJ	3.55	6.08	9.63	43.78	0.33	44.11	8.31	19.16	27.47
Shandong	SD	2.96	122.43	125.39	42.05	428.25	470.30	6.67	198.63	205.31
Guangdong	GD	2.44	33.67	36.12	33.67	74.68	108.35	5.99	55.05	61.05
Guangxi	GX	0.97	55.84	56.81	12.96	116.39	129.35	2.31	68.76	71.07
Hainan	HI	0.84	5.51	6.36	13.86	13.73	27.59	2.36	9.25	11.61

**Table 2 nutrients-14-04763-t002:** The daily food nutrient consumption per person in coastal areas of China.

Nutrients/Area ^1^	TJ	HE	LN	SH	JS	ZJ	FJ	SD	GD	GX	HI
Energy (kcal)	2156	1995	2248	2100	2084	2331	2172	1929	2265	2108	1814
Protein (g)	76	67	69	73	60	73	72	68	81	70	66
Fat (g)	77	63	78	79	71	86	80	64	91	76	73

^1^ TJ represents Tianjin, HE represents Hebei, LN represents Liaoning, SH represents Shanghai, JS represents Jiangsu, ZJ represents Zhejiang, FJ represents Fujian, SD represents Shandong, GD represents Guangdong, GX represents Guangxi, HI represents Hainan.

## Data Availability

Not applicable.

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
