# Peer review of "Spatial Differences of Nutrient Adequacy in Coastal Areas of China"

_nutrients, 2022, doi:10.3390/nu14224763_

Round 1

Reviewer 1 Report

Overall:

-          English language revisions and edits throughout need to be considered; a lot of errors with tenses of words, spelling of words, etc.

Abstract:

-          Line 11: ‘system’ should be ‘systems’

-          Be consistent with tense – use past tense in reference to all study activities

-          Seems to be lacking adequate detail on methods and analyses

Introduction:

-          Cite SDGs

-          The second paragraph is VERY long; consider breaking it up into smaller paragraphs

-          Is food security really a concern of academia? I am confused by this statement. It seems like an issue of society that researchers address

Methods/Results:

-          Why were only energy, fat, and protein explored? What about carbohydrates?

-          Also, it is unclear to me what was considered in the ‘protein’ category.  Were plant-based protein foods included?

-          Table 2: Please add a footnote with a key for the region abbreviations

-          It seems that Discussion was brought into the Results section a bit.  The Results section should just be presenting the data/numbers/results; discussion about inadequacies and government approaches should be left for the Discussion

Discussion:

-          Lines 376-387: The analyses did not include major food group, to my understanding, carbohydrates.  This should be discussed as a potential reason for the paradox observed as well. 

Author Response

Response to Reviewer 1 Comments

Dear editors and reviewers,

We sincerely thank the reviewers for providing constructive feedback, and help to improve the quality of the manuscript. We have reviewed the paper carefully, and modified the paper as per the reviewers’ comments. The manuscript is changed comprehensively to address the issue raised by the reviewers. All modifications used the "Track Changes" function. We hope our revised manuscript is suitable for the publication in Nutrients. Thank you very much for considering our revised manuscript.

Overall:

Comment 1: English language revisions and edits throughout need to be considered; a lot of errors with tenses of words, spelling of words, etc.

Response 1: Thanks for the comment. In order to accurately introduce the research content of this manuscript, we carefully read the full text and corrected some language and writing problems. We really hope that the language level has been substantially improved.

Abstract:

Comment 2: Line 11: ‘system’ should be ‘systems’

Response 2: Thank you. We have revised it as your suggestion. (Line 12)

Comment 3: Be consistent with tense – use past tense in reference to all study activities

Response 3: Many thanks. We accept this suggestion. We modified abstract to keep the tenses consistent and refer to all study activities in the past tense. (Lines 10-25)

Comment 4: Seems to be lacking adequate detail on methods and analyses

Response 4: We are grateful for this suggestion and we endorse it. We have added a description of the method and content so that authors can get a direct view of our study from the abstract. (Lines 14-17)

Introduction:

Comment 5: Cite SDGs

Response 5: Thanks for your advice. We have cited the literature on SDGs, i.e., [4] UN. Transforming our world: the 2030 Agenda for Sustainable Development. New York, 2015. (Line 34, Line 488)

Comment 6: The second paragraph is very long; consider breaking it up into smaller paragraphs

Response 6: We strongly agree with your comment and accept this suggestion. Paragraphs are too long making it more difficult to read, so we have divided them into 3 smaller paragraphs. (Lines 40-74)

Comment 7: Is food security really a concern of academia? I am confused by this statement. It seems like an issue of society that researchers address

Response 7: We appreciate your concern. Food security is a key concern in the academic field of agricultural policy, and is more evident in developing countries. Food security is, of course, closely linked to public nutrition security. Numerous scholars are involved in research that works to improve food nutrition security and reduce hunger and malnutrition around the world. At the same time, the issue has attracted the continued attention of international agencies, including the Food and Agriculture Organization of the United Nations, the International Fund for Agricultural Development, the World Food Programme, and the World Health Organization. A special issue of Nutrients, "Examining Linkages among Food Insecurity, Food Systems, and Public Health", also highlights the relationship between food and nutrition, and the relationship between food security and public health.

Methods/Results:

Comment 8: Why were only energy, fat, and protein explored? What about carbohydrates?

Response 8: Thank you for your comment. First, there are many types of nutrients, and limited by the difficulty of data collection, we selected 3 common and macronutrients of food, namely energy, protein and fat. Second, carbohydrates and energy have overlap, carbohydrates account for 50%-65% of energy in the nutritional intake of normal household members; therefore, we retained only energy. Third, in this paper, nutrients such as carbohydrates, dietary fiber, and vitamins were not considered, which is indeed a pity and we will refine them in further studies.

Comment 9: Also, it is unclear to me what was considered in the ‘protein’ category.  Were plant-based protein foods included?

Response 9: Yes. The protein in this paper is obtained from the transformation of animal food and plant food, so there are two kinds of protein, animal protein and plant protein. However, due to the setting of the research topic, we did not carry out an in-depth analysis of protein category.

Comment 10: Table 2: Please add a footnote with a key for the region abbreviations

Response 10: Thanks for the comment. The footnote for the region abbreviations has been added in page 8. In fact, region abbreviations are also shown in Table 1. The corresponding supplements are as follows.

TJ represents Tianjin, HE represents Hebei, LN represents Liaoning, SH represents Shanghai, JS represents Jiangsu, ZJ represents Zhejiang, FJ represents Fujian, SD represents Shandong, GD represents Guangdong, GX represents Guangxi, HI represents Hainan

Comment 11: It seems that Discussion was brought into the Results section a bit. The Results section should just be presenting the data/numbers/results; discussion about inadequacies and government approaches should be left for the Discussion

Response 11: Thanks for your comment. We have deleted the part about discussion in the Conclusion, and added data presentation. (Lines 40-471)

Discussion:

Comment 12: Lines 376-387: The analyses did not include major food group, to my understanding, carbohydrates. This should be discussed as a potential reason for the paradox observed as well.

Response 12: Many thanks. We agree with your comment. Indeed, the nutrient consumption analyses did not distinguish major food groups, and therefore, the analysis of potential reason related to the irrational intake of carbohydrates is added to the discussion of the fat paradox. (Lines 394-395)

Reviewer 2 Report

The purpose of this paper is to determine the nutrient adequacy of coastal provinces and spatial differences among provinces using Chinese government statistical data. Although it is a very interesting study, the following points may need to be revised. 1) If food were transported from states with sufficient food to states with insufficient food, no disparities would exit between coastal states. A more in-depth discussion of why disparities occur between coastal states would be needed. 2) Although it may be difficult due to data limitations, a longitudinal analysis is needed for further analysis. 3) The limitation of the study is not written.

Author Response

Dear editors and reviewers,

We sincerely thank the reviewers for providing constructive feedback, and help to improve the quality of the manuscript. We have reviewed the paper carefully, and modified the paper as per the reviewers’ comments. The manuscript is changed comprehensively to address the issue raised by the reviewers. All modifications used the "Track Changes" function. We hope our revised manuscript is suitable for the publication in Nutrients. Thank you very much for considering our revised manuscript.

The purpose of this paper is to determine the nutrient adequacy of coastal provinces and spatial differences among provinces using Chinese government statistical data. Although it is a very interesting study, the following points may need to be revised.

Response: Many thanks for your positive comment.

Comment 1: If food were transported from states with sufficient food to states with insufficient food, no disparities would exit between coastal states. A more in-depth discussion of why disparities occur between coastal states would be needed.

Response 1: Many thanks. We fully agree with the reviewers' comment. It is true that cross-regional trade and transport of food can make no differences in nutrient adequacy in coastal areas, but the difference is objective. The difference comes from the difference in food production endowment and difference in food production input elements. For this reason, we provide further discussion and explanation in lines 400-403.

Comment 2: Although it may be difficult due to data limitations, a longitudinal analysis is needed for further analysis.

Response 2: Yes, many longitudinal questions could not be analyzed in depth due to data limitations. We have added data presentation and summary of results by mining as many valuable and meaningful results as possible from the available data. At the same time, we also presented the research limitations. We will work further to add more nutrients, population groups, and time-series changes to the analysis in future studies to continuously enrich this series of research results. (Lines 243-245, Lines 248-249, Lines 430-439)

Comment 3: The limitation of the study is not written.

Response 3: We agree with you and accept this suggestion. We have added study limitation, mainly the selection of fewer nutrient indicators and the absence of nutrient adequacy analyses for different population groups. The modifications and supplements are as follows. (Lines 430-439)

Our research has several limitations. Due to data availability, only three nutrients were selected for this study: energy, protein, and fat. Many other nutrients still very important, such as carbohydrates, vitamins, and minerals, but were not included in the study. We encourage more scholars to participate in further research on the adequacy, balance and safety of more nutrients. This study focuses on spatial differences in nutrient adequacy, but neglects the nutritional status of different population groups, which is necessary and meaningful, especially focusing on low-income populations, populations in war-torn areas, women and children, etc. We strongly appeal local governments, international organizations, and social groups make efforts to help more people escape from nutrition insecurity.

Reviewer 3 Report

This paper presents a mathematical rationale for the nutritional adequacy of provinces located in coastal areas of China. It discusses the adequacy of nutrition through the ratio of nutrition production and nutrition consumption. 

It must be corrected.

table 2 Pat .. fat

Figure 3. Pat.. fat

Terminology needs to be unified.(agri-food.. agro-food.) 

Author Response

Response to Reviewer 3 Comments

Dear editors and reviewers,

We sincerely thank the reviewers for providing constructive feedback, and help to improve the quality of the manuscript. We have reviewed the paper carefully, and modified the paper as per the reviewers’ comments. The manuscript is changed comprehensively to address the issue raised by the reviewers. All modifications used the "Track Changes" function. We hope our revised manuscript is suitable for the publication in Nutrients. Thank you very much for considering our revised manuscript.

Comment 1: It must be corrected.

Table 2 Pat .. fat

Figure 3. Pat.. fat

Response 1: Thanks for pointing out the mistake. We have corrected them in Table 2 and Figure 3. We apologize for the errors mentioned above. In order to avoid such similar errors in this paper again, we have carefully checked the content of the whole paper, including figures and tables.

Comment 2: Terminology needs to be unified. (agri-food.. agro-food.) 

Response 2: Thanks for your comment, and we accept this suggestion. We use “agri-food” to unify terminology. (Line 379)

Round 2

Reviewer 1 Report

Thank you for addressing all of the comments.  If able, I still think a statement could be added that indicates what sources of protein are included in the protein category.  Are all sources of protein, including plant-based proteins, included? 

Author Response

Thank you very much for hearing from you again. We accept your comment and have completed the revision. The modification used the "Track Changes" function.

Comment: Thank you for addressing all of the comments. If able, I still think a statement could be added that indicates what sources of protein are included in the protein category.  Are all sources of protein, including plant-based proteins, included?

Response: Thanks for the comment. We agree with your comment and accept this suggestion. We have added a footnote on page 4 that clearly states the source of the protein in this paper and that it contains both plant-based and animal-based proteins.

Protein is derived from various types of food, therefore, protein in this paper includes not only plant-based proteins, but also animal-based proteins.